# Experimental observation of photonic nodal line degeneracies in metacrystals

Wenlong Gao[1], Biao Yang[1], Ben Tremain[2], Hongchao Liu[1], Qinghua Guo[1,3], Lingbo Xia[1,4], Alastair P. Hibbins[3] & Shuang Zhang[1]

Nodal line semimetals (NLS) are three-dimensional (3D) crystals that support band crossings in the form of one-dimensional rings in the Brillouin zone. In the presence of spin–orbit coupling or lowered crystal symmetry, NLS may transform into Dirac semimetals, Weyl semimetals, or 3D topological insulators. In the photonics context, despite the realization of topological phases, such as Chern insulators, topological insulators, Weyl, and Dirac degeneracies, no experimental demonstration of photonic nodal lines (NLs) has been reported so far. Here, we experimentally demonstrate NL degeneracies in microwave cut-wire metacrystals with engineered negative bulk plasma dispersion. Both the bulk and surface states of the NL metamaterial are observed through spatial Fourier transformations of the scanned near-field distributions. Furthermore, we theoretically show that the NL degeneracy can transform into two Weyl points when gyroelectric materials are incorporated into the metacrystal design. Our findings may inspire further advances in topological photonics.

[1] School of Physics and Astronomy, University of Birmingham, Birmingham B15 2TT, UK. [2] Department of Physics and Astronomy, Electromagnetic and Acoustic Materials Group, University of Exeter, Stocker Road, Exeter EX4 4QL, UK. [3] International Collaborative Laboratory of 2D Materials for Optoelectronics Science & Technology of Ministry of Education, Shenzhen University, Shenzhen 518060, China. [4] Center for Terahertz waves and College of Precision Instrument and Optoelectronics Engineering, Key Laboratory of Opto-electronics Information and Technical Science, Ministry of Education, Tianjin University, Tianjin 300072, China. Correspondence and requests for materials should be addressed to A.P.H. (email: A.P.Hibbins@exeter.ac.uk) or to S.Z. (email: s.zhang@bham.ac.uk)

Recent progress in the research on topological phases of matter has led to the exciting findings of Chern insulators[1], topological insulators[2,3], Weyl semimetals[4–6], Dirac semimetals[7], etc. The concept of nontrivial topological physics in solid-state systems has been extended to photonic systems, with particular interests focusing on achieving one-way disorder-immune surface states (SSs). Optical analogs of quantum Hall effect[8,9], quantum spin Hall effect[10–12], and valley Hall effect[13–16] have been realized in various two-dimensional (2D) photonic crystal systems. In three-dimensional (3D) photonic systems, there has been growing attention on Weyl degeneracies[17,18], 3D Dirac points[19], and photonic weak topological insulators[20]. One-way backscatter immune SSs, the so-called Fermi arcs, have been observed at the interface between a photonic Weyl system and a topologically trivial medium[21]. Among various topological photonic systems, metamaterials represent a unique effective medium approach for studying topological behaviors of electromagnetic waves, and have attracted growing research interest in recent years. For topological metamaterials, a simple homogenous model can greatly facilitate the investigation of important properties of the topological phases, and the topologically protected SSs are usually tightly confined to the interface due to the deep sub-wavelength unit cell of the structure[22]. Recently there has been realization of topological insulators[10–12,20], Weyl degeneracies[17,23], and Dirac degeneracies[24] in the metamaterial and metacrystal systems.

As a precursor of many novel topological phases, nodal line semimetals (NLS) have triggered a remarkable level of research interest since its first discovery in 2011[25]. NLS are topological semimetals with one-dimensional ring-shaped nodal line (NL) degeneracies in the 3D Brillouin zone[26]. In the condensed-matter systems, NLs have been recently found to exist in graphene networks[27], spin–orbit metal $PbTaSe_2$[28], alkaline earth metals[29], and a number of other material systems[30–32]. A salient feature of the NLS is that an eigenstate adiabatically transported along a closed loop threading the NL gains a $\pm\pi$ Berry phase, leading to a Zak Phase difference between the inside and outside of the ring[33]. Interestingly, NLS can transform into Weyl semimetals[34], Dirac semimetals[35], and 3D topological insulators[36] when spin–orbit coupling or other symmetry-lowering mechanisms are introduced. Recently, nodal-chains[37], nodal-links[38], and nodal-knot[39] semimetals are found to exist in solid-state systems. In contrast to the tremendous progress toward experimental demonstration of NLS in condensed-matter systems, there has been no report on experimental realization of NLs in the photonics regime[40].

In this communication, we report the experimental realization of a clean NL in a cut-wire metacrystal, which may provide a fertile ground for investigating not only the interesting topological features such as drumhead SSs and topological phase transitions between NL and Weyl degeneracies, but also a number of other interesting optical properties, for instance, resonance scattering[41] and negative refraction. Our study for the first time introduces NL degeneracies into the metacrystals and metamaterials, which may pave the way to exploration of the associated unusual optical phenomena.

## Results

**Design of the metacrystal.** Serving as one of the most basic building blocks of photonic metamaterials[42], cut-wire resonators provide a Lorentzian-shaped resonance in permittivity along the wire, which have been utilized to realize hyperbolic metamaterials[43], negative refractive index materials[44], and metasurfaces[45]. Noticeably, nonlocal effects are strong in wire-consisted metamaterials, which usually result in a positive dispersion for the longitudinal bulk plasmon (LP) mode along the wire[46]. Here, by introducing glide symmetry into the cut-wire metamaterial design, we realize a negative dispersion for the LP mode, which plays a key role in the formation of NL degeneracy in this work. The NL metacrystal is formed by stacking planar metallic cut-wire elements into a 3D array, as shown in Fig. 1a. The metacrystal unit cell, with a dimension of $4.5 \times 4.5 \times 2\ mm^3$, consists of two mutually orthogonal I-shaped metallic cut-wire resonators lying in the $x$–$y$ plane. The space group index of the metacrystal is P4/mbm (number 127), which exhibits two glide symmetries perpendicular to the main axis[47]. Comsol is used to simulate the photonic band structure of the metacrystal, with the results given in Fig. 1d. Along the in-plane directions, the lowest three bands are formed by two transverse modes—the transverse electric (TE) and the transverse magnetic (TM) modes, and a LP mode. The TE mode, having electric fields only in $z$-direction, exhibits a negligible interaction with the metallic cut wires, and therefore possesses a nearly linear dispersion of a large slope before reaching the Brillouin zone boundary. On the other hand, the TM mode strongly interacts with the cut wires in the $x$–$y$ plane, leading to a larger-effective index and consequently a smaller dispersion slope than the TE mode. The negatively dispersed LP mode, whose macroscopic electric field aligns with its wave vector, linearly intersects the TE mode at 16 GHz at points $U$ and $Y$ along $\Gamma$–$M$ and $\Gamma$–$X$ lines, respectively, serving as a strong indication of the presence of a degeneracy ring. The orthogonality between the LP and TE modes is guaranteed by the mirror symmetry of the system, and stays immune to any mirror symmetry preserved perturbations (Supplementary Note 6 and Supplementary Figure 6). The 3D-simulated band structure in the $x$–$y$ plane's Brillouin zone with $k_z = 0$ is shown in Fig. 1b. Note that the lowest band formed by the TM mode is not included in the plot for easy visualization. The 3D band structure confirms the ring degeneracy between the LP and TE modes. For non zero $k_z$, the ring degeneracy is gapped (Fig. 1c), further confirming the characteristics of a NL[9].

Besides the orthogonality between the LP and TE modes, the negative dispersion of the LP mode also plays a key role in the formation of a NL in the Brillouin zone. Here the negative dispersion is achieved by imposing the glide symmetry onto the metacrystal design, leading to the degeneracy between the LP and TM modes at the Brillouin edge. As shown in Fig. 1a, the structure is invariant under two-orthogonal glide symmetry operations: $\hat{G}_x : (x, y, z) \rightarrow (1/2 - x, 1/2 + y, z)$ and $\hat{G}_y : (x, y, z) \rightarrow (1/2 + x, 1/2 - y, z)$. In combination with the time-reversal symmetry operator $\Theta$, the new composite anti-unitary operators $\hat{G}_x\Theta$ and $\hat{G}_y\Theta$ guarantee doublet degeneracy for all bands along the $(k_{x,y} = \pm\pi, k_z = 0)$ high symmetry lines, which is analogous to the Kramer's pair for spinful electrons[48]. Here the TM and LP modes form a pair at the Brillouin zone boundary, as they can be transformed into each other by the in-plane group operations.

To confirm the existence of NL degeneracy, we numerically obtain the equi-frequency surface (EFS) at 16.325 GHz, which is slightly above the NL frequency (Fig. 1e). As expected, the EFS is in the form of a single torus in the whole Brillouin zone. It was reported recently that due to the diminishing surface area of the EFS at the Weyl frequency, a resonant scatterer embedded inside a photonic Weyl material could exhibit diverging resonance scattering cross-sections near the Weyl frequency[37] governed by an inverse square law. In contrast, the surface area of the EFS of a NL material has a linear dependence over the frequency close to the NL frequency (Supplementary Table 1). Thus, the NL metacrystal is expected to exhibit engineered electromagnetic scattering with scaling law different from that of Weyl materials.

A Hamiltonian formalism of the metacrystal is established to describe the dispersion close to the NL (Supplementary Note 1

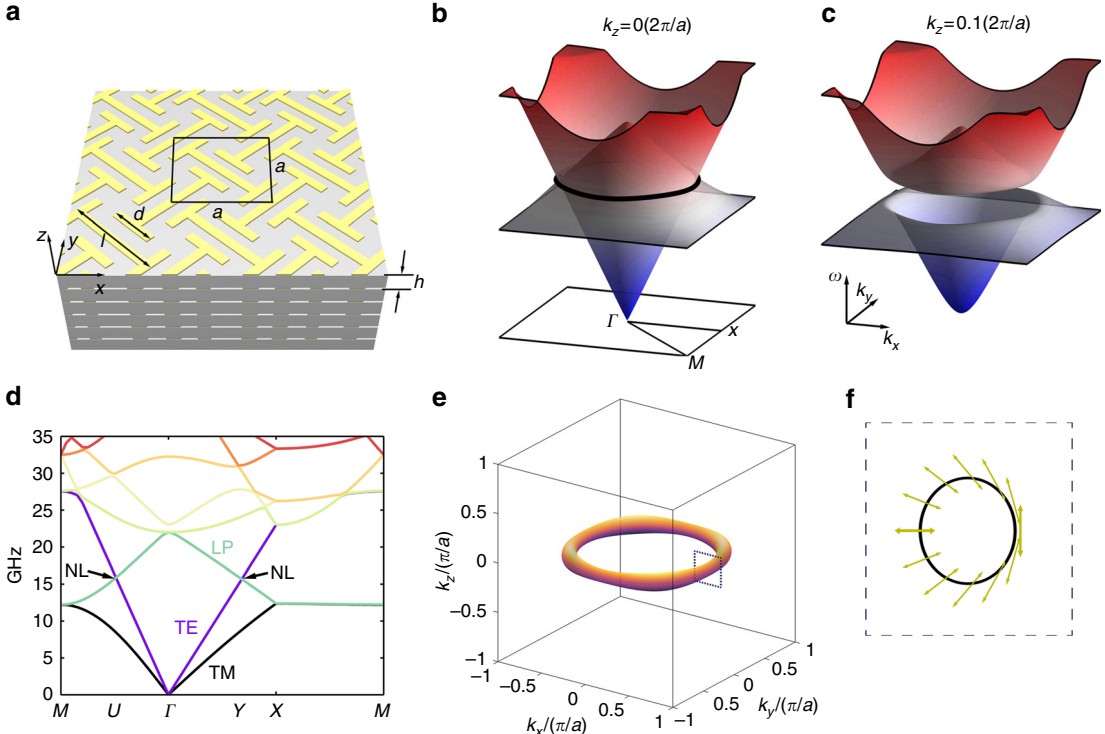

**Fig. 1** A single nodal line realized in cut-wire metacrystals. **a** Schematic of the metacrystal. The unit cell size is $a = 4.5$ mm, $h = 2$ mm. The size of the I-shaped copper cut-wire resonators is $l = 1.1a$, $d = 0.5a$. The widths of all wires are $0.1a$, and the thicknesses are 35 μm. The substrate material is Teflon, whose permittivity is 2.1, with a loss tangent around 0.00028. The band structure of the metacrystal at zero and non zero $k_z$ are given in **b** and **c**, respectively. **d** Band structure of the metacrystal at high symmetry lines. Nodal line (NL) is formed between the second and third band at 16 GHz at crystal momentum $U$ and $Y$. **e** The equi-frequency surface of the metacrystal exhibits a torus shape at 16.325 GHz, which is slightly above the NL frequency around 16 GHz. No other states are present in the Brillouin zone. **f** Polarization states on the NL calculated by an effective medium analysis. The area surrounded by the dashed line corresponds to the same area in **e**. Berry phase of the NL is well manifested by the nontrivial winding of polarization states

and Supplementary Figure 1). The electric field polarizations calculated by the Hamiltonian are given in Fig. 1f, serving as a manifestation of the $\pm\pi$ Berry phase of the NL. Specifically, going through a loop threading the NL on the EFS, the polarization state returns to the initial state but experiences a rotation of $\pi$. Interestingly, this polarization structure is reminiscent of that on a conical refraction ring surrounding the Dirac cone (Hamilton's Diablo) in biaxial birefringent crystals[49]. The $\pm\pi$ Berry phase through a loop threading the NL is confirmed numerically by using the Wilson-loop method[50] based on the Hamiltonian formalism of this strongly dispersive metacrystal[51] (Supplementary Note 3 and Supplementary Figure 3).

To account for the negatively dispersive LP mode, we empirically set the plasma frequency as a function of the in-plane momenta as $\omega_p = 1 - \alpha^2\left(k_x^2 + k_y^2\right)$, where $\alpha$ controls the slope of the LP mode. The effective Hamiltonian concerning only the LP and TE modes constituting the NL in vicinity to the NL frequency is expressed by

$$H = \sqrt{\frac{\omega_1^2}{2\varepsilon_s\eta}}\sigma_x k_z +$$
$$\left\{\sqrt{\frac{1}{\varepsilon_s}\left(1 + 2\alpha^2 k_{NL}\frac{\omega_1\omega_{NL}}{\eta}\right)}\sigma_z + \sqrt{\frac{1}{\varepsilon_s}\left(1 - 2\alpha^2 k_{NL}\frac{\omega_1\omega_{NL}}{\eta}\right)}I\right\}k_r,$$
$$(1)$$

where $\sigma_{x,z}$ are Pauli matrices, $\omega_0$ is the cut wires' resonance frequency, $\omega_1 = 1 - \alpha^2 k_{NL}^2$, $\eta = \omega_{NL}^2 + \omega_0^2 + \omega_1^2$, $\omega_{NL}$ is the NL's angular frequency, $k_{NL}$ is the NL's radial direction's momentum,

and $k_r$ is the momentum along the radial direction with respect to $k_{NL}$ (Supplementary Note 5 and Supplementary Figure 5). Obviously, this equation resembles 2D Dirac points with $\pi$ Berry phase, again confirming the existence of the NL and its Berry phase feature.

**Experimental observation of NL degeneracy.** To experimentally detect the NL degeneracy, a metacrystal is fabricated by using the standard printed circuit board technique. The fabricated sample is shown in Fig. 2a. A total number of 30 layers are stacked up to form the bulk metacrystal. Within each layer, there are 66 by 66 unit cells.

A microwave near-field scan system[27] is employed to detect both the bulk states and the SSs of the NL metacrystal. In the measurement of the bulk states (Supplementary Note 2), a $z$-polarized electric dipole is placed at the center of the bottom surface of the sample serving as the source, while another $z$-oriented dipole probe scans the top surface of the metacrystal to measure the $E_z$ field component. Evanescent tails of states in the bulk metacrystal can be collected by the probe dipole close to the surface. The source and probe dipoles are connected to a vector network analyzer (VNA) to measure both the magnitude and phase of the field. The real part of the measured instant field distribution in the real space at 16.325 GHz, which is slightly above the NL frequency, is shown in Fig. 2b. Circular wave fronts propagating along the radial direction are observed, indicating highly isotropic in-plane wave propagation. After Fourier transformation of the real space pattern, we obtain the projection of the EFS onto the surface Brillouin zone, which is given in Fig. 2c. The measured projected EFS exhibits an annular shape of

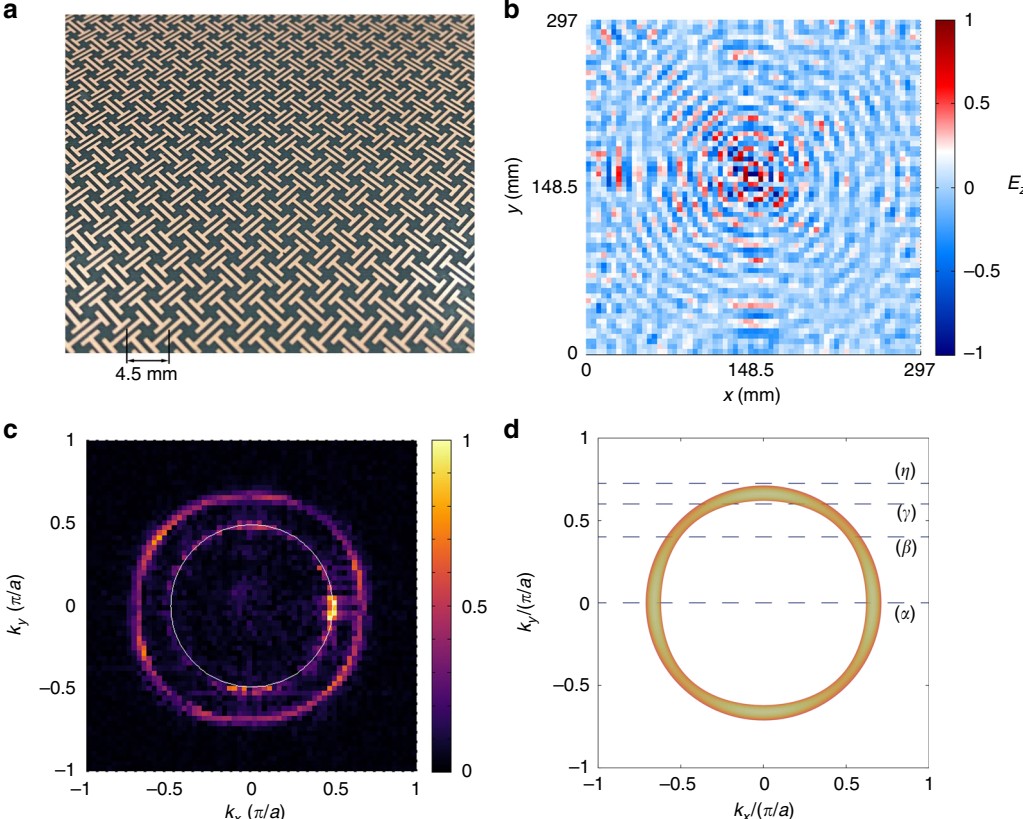

**Fig. 2** The measured and simulated equi-frequency contours. **a** The metacrystal is fabricated by the standard single layer printed circuit board technique. There are 66 unit cells in the lateral x- and y-direction and 30 in the z-direction, with each unit cell having a size $4.5 \times 4.5 \times 2$ mm³. **b** The measured distribution of the real part of the measured $E_z$ component (normalized to the maximum value) on the top surface of the metacrystal at 16.325 GHz. The source antenna is located at the center of the bottom surface. **c** Fourier transformed result of **b** exhibits a ring-shaped distribution. The thin white ring inside corresponds to the light cone. The Fourier components are normalized to the maximum. **d** Simulated EFS at 16.325 GHz. The dashed lines marked with $(\alpha),(\beta),(\gamma),$ and $(\eta)$ denote four different cuts at $k_y = 0.01, 0.4, 0.6,$ and $0.725$ $\pi/a$, respectively

finite width, which is consistent with the projection of a torus. For comparison, the numerically simulated EFS is given in Fig. 2d, which agrees reasonably well with the experimental result.

Furthermore, the spatial Fourier transformation is conducted over a broad frequency range between 12 and 20 GHz, which yields the information about the band structure of the metacrystal (Supplementary Note 2 and Supplementary Figure 2). Experimentally measured spatial Fourier components at the four different momentum cuts illustrated in Fig. 2d are presented in Fig. 3b, e, h, k. For the momentum cut intersecting the NL (lines α, β, and γ in Fig. 2d), cone-like dispersions are observed, with the cones' tips touching at around 16 GHz (Fig. 3b), confirming the linear band crossings forming the NL. The simulation results of a z-direction stacked 30-layers metacrystal are given in Fig. 3a, d, g, j (white dashed lines). The simulation and experimental results agree reasonably well—the simulated band structure overlaps well with the regions, where measured spatial Fourier components are strong. As shown in Fig. 3e, h for the momentum cut away from the Γ point (lines β and γ in Fig. 2d), the measured valleys broaden and shift toward the center. The measurement is in good agreement with the numerical result shown in Fig. 3d, h. When the momentum cut is further away from the Brillouin center (line η in Fig. 2d), the two valleys merge into each other forming a single valley in the middle (Fig. 3k). This is also well explained by the simulation result shown in Fig. 3j.

Another important signature of the NL degeneracy is the presence of the so-called drumhead SSs. In the near-field scanning measurement, SSs are also excited (marked in Fig. 3b,

e, h, k), due to scattering of bulk states at the edges of the metacrystal block. Numerically simulated dispersions of the SSs between the metacrystal and air along the four momentum cuts are presented in Fig. 3a, d, g and j. It is observed that the SSs are doubly degenerate at the Brillouin zone boundary, enforced by the glide symmetry. In order to achieve a better contrast of the SS to bulk states, a complementary measurement configuration is employed, wherein a y-polarized source dipole is placed close to the center of one edge of the top surface of the metacrystal, while the field on the top surface is scanned by a y-polarized scanning probe dipole (Supplementary Note 2). The measurement results are shown in Fig. 3c, f, i, l, showing good agreement with the simulation results. It should be noted that only SS with group velocity of negative sign is collected by the probe dipole as the source is located at the rightmost edge as shown in Fig. 1a.

## Discussion

Through numerical studies, we show that by incorporating a gyroelectric material into the metacrystal structure, the NL is gapped everywhere except at two discrete points along the applied magnetic field, which are identified as Weyl points (WPs). We apply the most well-known gyroelectric material, magnetized plasma to break the time-reversal symmetry. It has been reported previously that a magnetized plasma itself possesses WPs at its plasma frequency[41]. However, generation of WPs in magnetized plasma usually requires a very strong applied magnetic field, and the system is not clean in the sense that there exist extra bulk

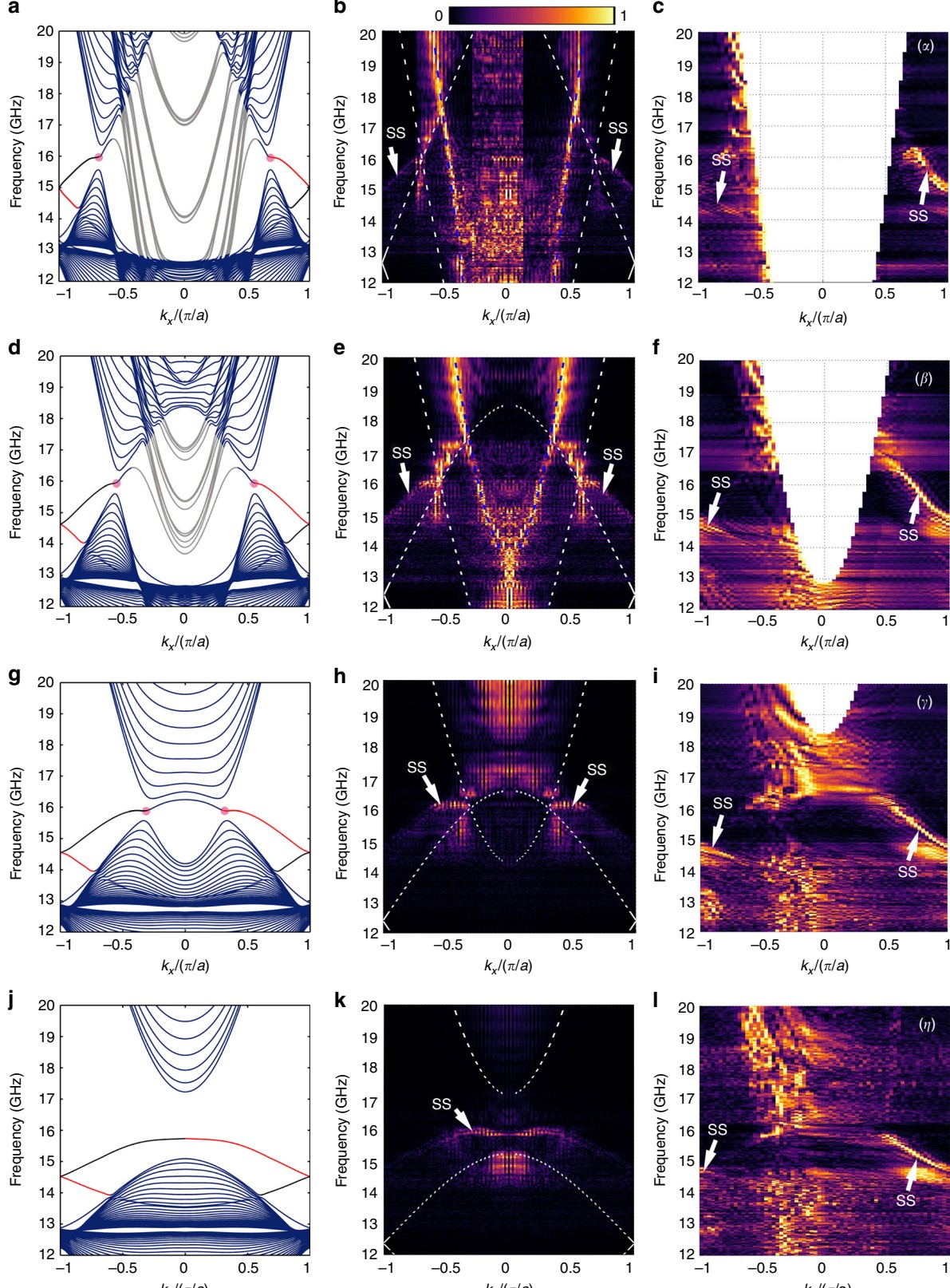

**Fig. 3** Band structure of the nodal line metacrystal. **a**–**c** Simulated and measured band structure of the 30-layers metacrystal along the ($\alpha$) line. In **a** the NL is denoted by the magenta dots in the two valleys. Surface states (SSs) are colored in red and black according to the sign of their group velocities. **b** Band structure extracted from the $E_z$ field measured on the top surface. The cone-shaped valleys touch at around 16 GHz, indicating the bulk states of the NL. SSs are also excited (indicated by the white arrows) through scattering of bulk states at the sample's edge. **c** SSs measured by placing a $y$-polarized source dipole close to the center of one edge of the top surface of the metacrystal. Note that only the red branch SSs in **a** are observed. Results within light cone are excluded for better visualization of the SSs. In **a** and **b**, the gray-colored bands correspond to extra waveguides modes in air which are not at present in the experiments. Results on ($\beta$), ($\gamma$), and ($\eta$) lines are given in **d**–**f**, **g**–**i**, and **j**–**l**

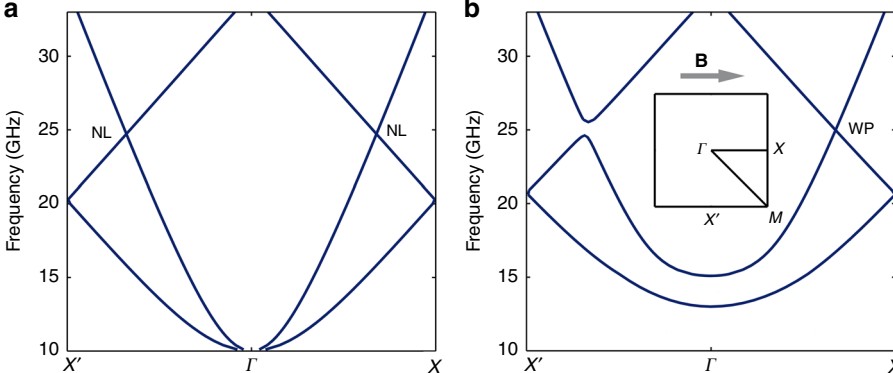

**Fig. 4** Transformation from nodal line to Weyl points. **a** When only plasma is applied to the metacrystal without a biased magnetic field, the nodal line (NL) is still present, but blue-shifted to 25 GHz. The plasma frequency is 10 GHz, rendering a photonic band gap at lower frequencies. **b** A magnetic field is applied parallel to the $\Gamma$–$X$ direction, as shown in the inset. This configuration leaves band crossings of the NL on the $\Gamma$–$X$ high symmetry line still closed, but gapped elsewhere. Hence two type-I Weyl points are formed in the Brillouin zone

states at the WP frequency. Here we show that two type-I WPs of opposite chirality can result from the combination of a NL with magnetized plasma, while the required plasma frequency and cyclotron frequency can be significantly lower than the WP frequency.

Here realistic frequency-dependent material parameters are used in our modeling. In the new configuration, the array of cut wires, instead of being attached to dielectric substrates, are submerged into a magnetized plasma with homogeneous electron density of $1.24 \times 10^{12}$ cm$^{-3}$ under a 0.357 T static magnetic field in the $x$ direction. The corresponding plasma frequency and cyclotron frequency are both around 10 GHz[52].

For comparison, the band structure of the hybrid metacrystal without an applied magnetic field is given in Fig. 4a. The plasma's permittivity at infinite frequency is 1, which is much smaller than the previously used supporting substrate. Consequently, the band structure is blue-shifted, which raises the NL frequency to about 25 GHz. Without the magnetic field, MP shows a simple Drude response in permittivity, and therefore the band structure is gapped below 10 GHz, due to the metallic response of MP below its plasma frequency.

Once the magnetic field is applied, the NL is transformed into two WPs. As shown in Fig. 4b, band crossing only persists along the direction of the magnetic field, forming a type-I WP, and is gapped elsewhere. This feature is well captured by the Hamiltonian formalism (Supplementary Note 4 and Supplementary Figure 4), and the effective Hamiltonian of the WP is expressed as

$$H = \nu_1 \sigma_x k_z + (\nu_2 \sigma_z + \nu_3 I)k_x + \nu_4 \sigma_y k_y, \qquad (2)$$

where $\nu_{1,2,3}$ can be expressed in terms of the field components of the eigen fields at the NL. When there is no static magnetic field, Eq. (2) recovers to Eq. (1) (Supplementary Note 5). Along any other directions, our calculation confirms that the NL is gapped, leaving only two point-like EFS at the WP frequency.

To conclude, we experimentally demonstrate a metacrystal possessing an optical NL in the form of a single ring. By introducing glide symmetry into the metacrystal design, we can engineer the nonlocality of the longitudinal bulk plasma mode to exhibit a negative dispersion. This unique feature gives rise to a single NL in the whole Brillouin zone. Our experiments verify the presence of both the toroidal bulk state and the drumhead SS supported by the metacrystal. We further numerically show that the NL can transform into two type-I WPs under a static magnetic field. Besides the unique topological characteristics, the NL degeneracy and donut-shaped EFS may introduce interesting

phenomena that arise from the unique optical density of states of the NL optical material, such as spontaneous emission, resonant scattering, and black-body radiation.

**Data availability**. The data that support the findings of this study are available from the corresponding author upon request.

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

## Acknowledgements

This work was financially supported by the European Research Council Consolidator Grant (TOPOLOGICAL). S.Z. acknowledges support from the Royal Society, the Wolfson Foundation, Horizon 2020 Action Project No. 734578 (D-SPA), the Leverhulme Trust (RPG-2012-674), and the Engineering and Physical Sciences Research Council (EP/J018473/1). B.Y. acknowledges support from the China Scholarship Council (201306110041). A.P.H. acknowledges financial support from EPSRC of the United Kingdom (Grant No. EP/L015331/1). Near-field scanning data were collected by VNA controlled with xyz-stage at G31 at the Department of Physics and Astronomy, University of Exeter, United Kingdom.

## Author contributions

W.G. and S.Z. conceived the idea, W.G., B.Y., B.T. and Q.G. designed and performed the experiments. W.G., B.T., A.P.H., and S.Z. analyzed the data. W.G., H.L., L.X. and S.Z. wrote the manuscript. All authors contributed to the discussion of the project.

## Additional information

**Competing interests:** The authors declare no competing interests.

