## [Peer Review File · Nature Communications]

Reviewers' comments:

Reviewer #1 (Remarks to the Author):

The main claim of the paper is the first experimental realization of a photonic nodal line metamaterial. As a generalization of the 2D Dirac points to 3D systems, nodal lines have attracted tremendous theoretical interest in the past few years, but they have been challenging to demonstrate in experiment. For example, while the first theoretical proposal came in 2011, they have only very recently been observed in condensed matter (arXiv:1708.06874) and I am not aware of any photonic realizations to date. Thus, the present work may be an important milestone and attract broad interest by demonstrating this class of systems in photonics for the first time. However, I do not think the authors have provided sufficient evidence to verify that their structure indeed hosts a nodal line.

The authors have considered a Hermitian effective medium, neglecting the effect of losses in their structure. This approach can only be justified if the losses are uniform to a good approximation and do not affect the modes hosting the nodal ring in different ways. This does not appear to be the case in the experimental observations: the states below the claimed nodal line in Fig. 3(b,e,h) are much more strongly attenuated than those above. This makes me question whether the structure truly exhibits a nodal ring degeneracy, or merely a crossing in the real part of the eigenvalues (while their imaginary parts remain nondegenerate).

I also note that Ref. [24] (from 2013) already presented a photonic crystal design hosting nodal loops based on an all-dielectric structure that does not suffer from losses like the present design (this design was however not implemented in experiment). The existence of a nodal ring was rigorously established there from first principles calculations. Unfortunately the nonlocal modal required by the authors here is only empirical and cannot be derived microscopically, and therefore the presence of a true nodal ring cannot be rigorously proven.

If a microscopic model for the authors' structure cannot be obtained, I think at the very least the authors need to provide a simulation of the scattering spectra of Fig. (b,e,h) based on their effective Hamiltonian to verify that it correctly describes their structure and determine whether the sharp cutoff in the measured band structure can be explained by uniform losses that do not lift the nodal line degeneracy.

Reviewer #2 (Remarks to the Author):

The authors propose and study numerically/experimentally a metamaterial with a photonic nodal ring degeneracy for in-plane propagation. The design is based on a cut-wire metamaterial with a nonlocal response. The authors show that for a particular frequency the TM and longitudinal mode are degenerate on a ring (due to a clever design that exploits a glide-symmetry to ensure the existence of Kramer's doublets at the corners of the BZ), and that the bulk states have a toroidal-type equifrequency surfaces.

The authors claim that this design is topological. However, it is rather unclear from the text what is exactly the topological property of the design. Topological systems should have some property that stays invariant under continuous deformations. What is exactly the property of the present material that stays invariant if its parameters are continuously varied? Is it the nodal line? If so, it seems that the result will only hold if the glide symmetry is preserved. This needs to be clarified.

Moreover, it is unclear from the article if this material with a nodal ring has any application. For example, does it support protected states? I think some new effect (beyond some exotic equifrequency surface) is essential to justify publication in a high impact journal.

Reviewer #3 (Remarks to the Author):

The authors propose a concept of optical nodal line in the whole Brillouin zone, realized by a periodic-lattice structure with an array of cut-wire metamaterials. It is novel for the field of topological photonics but this physics is similar to condensed matter. They also show the microwave observation of photonic bands using the experiment technique in Fourier transformation of near-field patterns, but the current version is hard to convince me. This work facilitates the research on topological phases of optical materials. More comments are as followed:

1. Nodal lines have been experimentally realized in many condensed matter systems. In order to arise broad interest, the authors should additionally state the advantages of the presented work in photonic system over condensed matter works or previously theoretical works in photonic systems.

Otherwise, this work will give a false impression that it is only a quantum simulation of nodal line in optical version.

2. The authors should seriously consider whether the system in this work belongs to metamaterial or photonic crystal. The widely accepted properties of metamaterials are determined by the responses of individual sub-wavelength structures, so that they can be considered as homogeneous media characterized by macroscopic material parameters. However, in this paper, the bands in the Brillouin zone are derived from Bragg diffraction in photonic crystals. Although the cut-wire metamaterials in this work are applied to construct the unit cell, the intrinsic mechanism should be attributed to Bloch waves of photonic crystals. Therefore, I think the system should more precisely refer to plasmonic crystal or photonic metacrystal rather than the term photonic metamaterial.

3. One of the key points in this work is to experimentally realize photonic nodal line. But the experimental method in the text is not enough to support the research. The authors should give more details about the microwave experiment. For example,

(1) Details about experimental setup, including model of vector network analyser and resolution of near-field scanning system in real space.

(2) It is not easy to determine the experimental results match well with simulations, just as Fig 3. More details about how to extract the band structures from measured field patterns should be shown in supplemental materials.

(3) In experiment, how to selectively excite the eigen states at a certain k_y plane?

4. In Fig 4, the nodal line will spawn into Weyl points when adding external magnetic field. Except for band structures, I recommend the authors to develop some new physics charts to compare with the intrinsic mechanism between nodal line and Weyl point, such as propagating properties and topological transitions.

5. In order to observe the photonic states below the nodal line in experiment, could the metamaterial resonance frequency tune to deviate from the operating frequencies?

6. It had better to label the indices of bands in Fig. 1(d) or label with high-contrast color, as it fails to recognize the 2nd and 3rd bands.

7. There are some grammatical errors that should be corrected in the manuscript, such as the spelling of Brillouin.

Reviewers' comments:

Reviewer #1 (Remarks to the Author):

The main claim of the paper is the first experimental realization of a photonic nodal line metamaterial. As a generalization of the 2D Dirac points to 3D systems, nodal lines have attracted tremendous theoretical interest in the past few years, but they have been challenging to demonstrate in experiment. For example, while the first theoretical proposal came in 2011, they have only very recently been observed in condensed matter (arXiv:1708.06874) and I am not aware of any photonic realizations to date. Thus, the present work may be an important milestone and attract broad interest by demonstrating this class of systems in photonics for the first time. However, I do not think the authors have provided sufficient evidence to verify that their structure indeed hosts a nodal line.

The authors have considered a Hermitian effective medium, neglecting the effect of losses in their structure. This approach can only be justified if the losses are uniform to a good approximation and do not affect the modes hosting the nodal ring in different ways. This does not appear to be the case in the experimental observations: the states below the claimed nodal line in Fig. 3(b,e,h) are much more strongly attenuated than those above. This makes me question whether the structure truly exhibits a nodal ring degeneracy, or merely a crossing in the real part of the eigenvalues (while their imaginary parts remain nondegenerate).

Reply:

We thank reviewer for considering the topic to be interesting and important. The challenge in observing the lower half of the nodal line was caused by two factors: the loss of the dielectric substrate and the proximity of the nodal line to the resonance of the structure. Note that in the microwave regime, the metal nearly functions as a perfect electric conductor with negligible loss, while the dielectric substrate material, FR4, has quite significant loss at the operating frequencies around 10 GHz. Therefore, the intrinsic loss arises primarily from the dielectric substrate. In the revised manuscript, we present new results on a different sample made on a dielectric substrate with lower dielectric constant, and much lower loss. The new sample allowed observation of both the upper and lower parts of the nodal line dispersion.

I also note that Ref. [24] (from 2013) already presented a photonic crystal design hosting nodal loops based on an all-dielectric structure that does not suffer from losses like the present design (this design was however not implemented in experiment). The existence of a nodal ring was rigorously established there from first principles calculations. Unfortunately the nonlocal modal required by the authors here is only empirical and cannot be derived microscopically, and therefore the presence of a true nodal ring cannot be rigorously proven.

Reply:

The presence of nodal line is justified in the first principle studies by finite element method simulations of Maxwell equations in COMSOL. The existence of the nodal line can be rigorously proven by inspecting the eigen-states' σ_z mirror reflection symmetry. The nodal line is formed by the interception of the longitudinal plasma (LP) mode and the TE mode. Electric field distributions on x-z plane of the two modes at Bloch wave vector: $\vec{k} = \frac{\pi}{a}(0.5, 0, 0)$ are given in the pictures below (**a** for LP and **b** for TE mode). They evidently exhibit σ_z eigen value 1 and -1 respectively, which guarantees the crossing of the two modes.

If a microscopic model for the authors' structure cannot be obtained, I think at the very least the authors need to provide a simulation of the scattering spectra of Fig. (b,e,h) based on their effective Hamiltonian to verify that it correctly describes their structure and determine whether the sharp cutoff in the measured band structure can be explained by uniform losses that do not lift the nodal line degeneracy.

Reply:

To observe the lower band of the NL, we fabricated another metamaterial using a low loss substrate material, Teflon, whose loss tangent is 0.00028, while loss tangent of the FR4 board used in our previous metamaterial design is 0.008. Since the real part of Teflon's permittivity is 2.1, the NL is blue-shifted to 16 GHz, while the resonance frequency increases to 13 GHz. The separation between the NL and the resonance frequencies is 3 GHz, which is twice that of the FR4 sample (11GHz - 9.5GHz = 1.5 GHz). The measured spectra at various momentum cut in the Brillouin zone together with the simulation results are given in the following picture.

In the above figure, panels α , β , γ and η represent different momentum cuts at $k_y = 0.01, 0.4, 0.6$ and $0.725 \pi/a$, respectively. The lower band of NL is clearly observed in the low loss metamaterial design. The white dashed lines denote the corresponding simulated band structures at each momentum cut with $k_z = 0$.

In the simulation we discover that the nodal line degeneracies for both the real part and the imaginary part of the dispersion occur at the same momentum, and their presence is very robust against material loss. In the microwave regime, metals can be well approximated by Perfect Electric Conductors(PEC). However, the loss tangent of the dielectric substrate could be important. As is shown in the following picture, the band structure is simulated in COMSOL by including the loss parameters. We first consider the loss only from the dielectric substrate. Here loss tangent of Teflon is exaggerated to be 0.2, which is more than 600 times of the realistic value. Simulation shows that the real and the imaginary parts of the nodal line are simultaneously degenerate at the NL momentum, as is shown in the picture below. Next, a finite conductivity (6×10^7 S/m for copper) is assigned to the metal, which, however, results in negligible influence on NL, as shown by the figure below. Therefore the loss should not cause the lifting of the NL degeneracy.

Reviewer #2 (Remarks to the Author):

The authors propose and study numerically/experimentally a metamaterial with a photonic nodal ring degeneracy for in-plane propagation. The design is based on a cut-wire metamaterial with a nonlocal response. The authors show that for a particular frequency the TM and longitudinal mode are degenerate on a ring (due to a clever design that exploits a glide-symmetry to ensure the existence of Kramer's doublets at the corners of the BZ), and that the bulk states have a toroidal-type equifrequency surfaces.

The authors claim that this design is topological. However, it is rather unclear from the text what is exactly the topological property of the design. Topological systems should have some property that stays invariant under continuous deformations. What is exactly the property of the present material that stays invariant if its parameters are continuously varied? Is it the nodal line? If so, it seems that the result will only hold if the glide symmetry is preserved. This needs to be clarified.

Reply:

We thank reviewer for the comments. The NL is formed by the degeneracy between the longitudinal mode and the transverse electric mode. The existence of the NL is protected by the mirror symmetry instead of the glide mirror symmetry. Here the glide symmetry is utilized to introduce degeneracy between the LP mode and the TM mode at the Brillouin zone edge, ensuring a negative dispersion for the LP mode and a clean nodal line in the momentum space free of other bulk modes at the same frequency. Breaking of the glide symmetry does not lift the degeneracy, which is illustrated by the band structure of the metamaterial with broken glide symmetry by shrinking the size of one of the cut wire resonator ($l_2=0.8 \cdot l_1$) in following picture:

Note that although the doubly degeneracy at Brillouin zone boundary enabled by the glide reflection symmetry is lifted, the NL degeneracy persists. However, in this configuration the band structure close to the nodal line frequency is not clean as that with glide mirror symmetry (Fig. 1d of the main text). Nevertheless, any deformations apart from the mirror symmetry breaking will not lift the NL degeneracy.

Moreover, it is unclear from the article if this material with a nodal ring has any application. For example, does it support protected states? I think some new effect (beyond some exotic equifrequency surface) is essential to justify publication in a high impact journal.

Reply:

The nodal line metamaterial does support surface states, as is shown in Fig. 3 in the main text. However, surface states of nodal line semimetal are not topologically protected, and could merge into bulk bands by tuning surface termination [Lu, L & et al. Nat. Photon. 7, 294–299 (2013)]. On the other hand, the NL degeneracy and donut-shaped isosurface may introduce new interesting phenomena that arise from the unique optical density of states of the nodal line optical material, such as spontaneous emission, resonant scattering, black body radiation. In a recent paper [M. Zhou & et. al. Nature Communications 8, 1388 (2017)], it was shown that a resonant scatterer embedded inside a Weyl photonic material exhibits diverging resonance scattering cross sections near the Weyl frequency, and the resonance scattering cross section obeys a scaling law $\sigma \sim 1/S \sim 1/(\omega_{\text{weyl}} - \omega_0)^2$, where S is the surface area of the equi-frequency surface, ω_0 is the resonance frequency of the scatterer, and ω_{weyl} is the Weyl frequency.

In contrast, the equi-frequency surface of a nodal line material is toroidal, whose surface area can be expressed as $S = 2\pi k_{\text{NL}} k_r \sim k_{\text{NL}} (\omega_{\text{NL}} - \omega_0)$ where k_{NL} is distance from Brillouin zone center to center of the torus' tube, and k_r is the radius of the tube. Thus resonant scattering inside NL is expected to exhibit a linear scaling law for the scattering cross section, i.e. $\sigma \sim 1/(\omega_{\text{weyl}} - \omega_0)$.

Another possible application of the nodal line metamaterial could be negative refraction. The conventional designs of negative index materials are complex in design, requiring both

electrically and magnetically resonance structures. In contrast, the nodal line metacrystal only consists of electric dipole resonators. This may inspire further studies on interesting optical properties of negative refractions based on this platform. For instance, negative refraction could take place between two NL materials with different NL frequency, as is illustrated in the picture below, where the black lines are cut of the equi-frequency-surface of NL material, and the red arrows indicate group velocity direction.

Reviewer #3 (Remarks to the Author):

The authors propose a concept of optical nodal line in the whole Brillouin zone, realized by a periodic-lattice structure with an array of cut-wire metamaterials. It is novel for the field of topological photonics but this physics is similar to condensed matter. They also show the microwave observation of photonic bands using the experiment technique in Fourier transformation of near-field patterns, but the current version is hard to convince me. This work facilitates the research on topological phases of optical materials. More comments are as followed:

Reply: We thank the reviewer for the comments and for considering our work to be novel in the field of topological photonics. In the revised manuscript, we present new experimental results on a different sample made on a dielectric substrate with lower dielectric constant, and a much lower loss. The new sample allows experimental observation of both the upper and lower parts of the nodal line dispersion.

1. Nodal lines have been experimentally realized in many condensed matter systems. In order to arise broad interest, the authors should additionally state the advantages of the presented work in photonic system over condensed matter works or previously theoretical works in photonic systems. Otherwise, this work will give a false impression that it is only a quantum simulation of nodal line in optical version.

Reply:

Besides the fundamental interest finding NL in metamaterial/ metacrystal systems, the photonic NL metacrystal in our design may host a number of phenomena that are unique for optical systems. In particular, the NL degeneracy and donut-shaped isosurface may introduce new interesting phenomena that arise from the unique optical density of states of the nodal line optical material, such as spontaneous emission, resonant scattering, and black body radiation. It should be noted that the NL demonstrated in our work is ideal, meaning that no other bulk states are present in the whole Brillouin zone around the NL frequency. This unique feature, to the best of our knowledge, has not been realized in condensed matter systems. This advantage could facilitate studies of the scattering properties of resonators

embedded in the NL metamaterial without the interference from other bulk states. For instance, the Weyl photonic material was found to possess diverging resonance scattering cross sections near the Weyl frequency, with the scaling law obeying $\sigma \sim 1/S \sim 1/(\omega_{\text{weyl}} - \omega_0)^2$ [M. Zhou & et. al. Nature Communications 8, 1388 (2017)]. In contrast, NL is expected to exhibit a linear scaling law of resonance scattering cross section $\sigma \sim 1/(\omega_{\text{weyl}} - \omega_0)$ due to the linear dependence of density of states over frequency. However, in a not clean system, the presence of extra bulk states in the Brillouin would invalidate the divergence feature of scattering cross sections.

The nodal line metacrystal could also be used to study negative refraction. Usually, negative index materials are complex in design, requiring both electrically and magnetically resonance structures. In contrast, the nodal line metacrystal only consists of electric dipole resonators. For instance, negative refraction could take place between two NL materials with different NL frequency, as is illustrated in the picture below, where the black lines are cut of the equi-frequency-surface of NL material, and the red arrows indicate group velocity direction.

Finally, our work is the first experimental realization of a clean nodal line degeneracy in photonic systems. While there have been some theoretical proposals [Lu, L & et al. Nat. Photon. 7, 294–299 (2013)] on photonic crystal approach towards photonic nodal lines, there has been no experimental demonstration. Furthermore, in comparison to the photonic crystal approach, our design benefit from the sub wavelength scale of the metacrystal unit cell in two ways. First, the NL is located well outside the light cone, leading to better localization of the surface states. Second, a homogenized field description could be assigned to the metacrystal. This simplicity of the design may inspire further studies based on this platform.

2. The authors should seriously consider whether the system in this work belongs to metamaterial or photonic crystal. The widely accepted properties of metamaterials are determined by the responses of individual sub-wavelength structures, so that they can be considered as homogeneous media characterized by macroscopic material parameters. However, in this paper, the bands in the Brillouin zone are derived from Bragg diffraction in photonic crystals. Although the cut-wire metamaterials in this work are applied to construct the unit cell, the intrinsic mechanism should be attributed to Bloch waves of photonic crystals. Therefore, I think the system should more precisely refer to plasmonic crystal or photonic metacrystal rather than the term photonic metamaterial.

Reply:

We thank the reviewer for the comment. We agree with the reviewer that nonlocality is present in the material and indeed nonlocality plays a key role for achieving the negative dispersion for the longitudinal plasmon mode. Accordingly, the title of our manuscript has been changed to ‘Experimental observation of photonic nodal line degeneracies in metacrystals’.

3. One of the key points in this work is to experimentally realize photonic nodal line. But the experimental method in the text is not enough to support the research. The authors should give more details about the microwave experiment. For example,
 (1) Details about experimental setup, including model of vector network analyser and resolution of near-field scanning system in real space.

Reply:

We thank the reviewer for the comment. More detailed information of the experimental setup has been added to section 2 ‘Experimental setup’ of the supplementary information.

(2) It is not easy to determine the experimental results match well with simulations, just as Fig 3. More details about how to extract the band structures from measured field patterns should be shown in supplemental materials.

Reply: We thank the reviewer for the comment. Briefly, we have carried out measurement of the near field distribution by scanning a probe on the top surface of the sample. Since the wave vector at the nodal line are outside the light cone, the modes decay away from the top surface into the air. By using near field scanning we are able to measure the evanescent wave on the air side. By performing Fourier transformation of the near field pattern, we obtain the projection of the isosurface onto the x-y plane at each frequency, which contains the band structure information. In what follows, we give more details on the correspondence between the measured spectrum and the simulated one.

As is shown in (a) of the above figure, when a bulk state in the metacrystal is excited from the bottom surface of the sample and propagates to the top interface, it experiences total internal reflection and is reflected back into the metacrystal. The evanescent tail in the air can be detected by the near field antenna. (c) illustrate an example of equi-frequency contour. In the experiment, since the metacrystal has finite thickness in the z direction, the allowed kz is discretized, as is illustrated in (d), where each loop represents a cut of the EFC in (c) at a constant kz. Since the fields are only detected at the upper boundary, the Fourier transformation of the field represents the projection of the bulk isosurface, i.e. a set of rings

(d), onto the k_x - k_y plane (b). This explanation is added into section 2 ‘Experimental setup’ of the supplementary information.

(3) *In experiment, how to selectively excite the eigen states at a certain k_y plane?*

Reply: In our experiment, we used a point source which can excite a broad range of wave vectors. The Fourier transformation of the field pattern on the top surface (x - y plane) at each frequency can provide the projected dispersion along any in-plane directions. Nevertheless, the reviewer raised a very interesting question in the experiment perspective. We propose two possible methods for exciting selectively the modes with fixed k_y , as is illustrated in the following picture:

Picture **a** describes a configuration in which a line defect along y direction is placed between a high refractive index material (a prism) and the metacrystal. Incident light with a definite k_y determined by the incident angle could be scattered by the line defect, while keeping k_y a conserved parameter. However this method would need delicate design of the scattering cross section of the defect to efficiently excite light into the eigen-states of the metacrystal. An alternative method is shown in picture **b**. A periodic array of dipole antennas with a constant phase difference between neighboring elements are placed on top of the metacrystal. The excited eigen states thus satisfy $k_y = \frac{2\pi}{3a}$, where a is the spacing of the antennas.

4. *In Fig 4, the nodal line will spawn into Weyl points when adding external magnetic field. Except for band structures, I recommend the authors to develop some new physics charts to compare with the intrinsic mechanism between nodal line and Weyl point, such as propagating properties and topological transitions.*

Reply: We thank the review for the comment. A table summarizing some properties of the nodal lines and Weyl points is shown below, and has been added to as the last part of the supplementary material. By introducing external magnetic field, the nodal lines can experience topological transition and become Weyl points. If the σ_z mirror symmetry is broken, the nodal lines will transform into a normal insulator with a trivial 3D photonic band gap. However, the Weyl points are topologically stable as source or drain as Berry curvature, and can't not be lifted by any symmetry lowering.

	Nodal lines	Weyl points
Dimension	1D	0D
Berry curvature	No	Yes
Density of states	$ \omega - \omega_{NL} $	$ \omega - \omega_{Weyl} ^2$
Stability	stabilized by σ_z	stable under any perturbation
Transition	Weyl points/ Normal Insulator	N/A

5. In order to observe the photonic states below the nodal line in experiment, could the metamaterial resonance frequency tune to deviate from the operating frequencies?

Reply:

We thank the reviewer for the comment, we have fabricated another metacrystal with a different substrate, Teflon, with a lower dielectric constant (2.1) and much lower loss tangent of 0.00028. The NL frequency is 16GHz, and the resonance frequency is 13GHz. Experiment results are given in the following picture, and have been summarized in Fig. 3 in the main text. Lower bands of the NL can thus be measured. Panels α , β , γ and η represent different momentum cuts at $k_y = 0.01, 0.4, 0.6$ and $0.725 \pi/a$, respectively. The lower band of NL is clearly observed in the low loss metamaterial design. The white dashed lines denote the corresponding simulated band structures at each momentum cut with $k_z = 0$.

6. It had better to label the indices of bands in Fig. 1(d) or label with high-contrast color, as it fails to recognize the 2nd and 3rd bands.

Reply:

Following the suggestion we have changed the indices of bands to colors with better contrast.

7. There are some grammatical errors that should be corrected in the manuscript, such as the spelling of Brillouin

Reply:

We thank the reviewer for the comment, we have checked and corrected the grammatical errors in the manuscript.

REVIEWERS' COMMENTS:

Reviewer #1 (Remarks to the Author):

I thank the authors for their comprehensive response. The finite element simulations of the eigenmodes convincingly demonstrate that the nodal-line degeneracy is not lifted by losses, and the experimental results are clearer with the use of the new lower loss structure. Therefore I think the major claims of the paper are now adequately supported. Before publication the authors should carefully check the text to ensure details are updated to reflect the new structure, e.g. line 101 refers to the previous nodal line frequency of 12.5GHz.

Reviewer #2 (Remarks to the Author):

I thank the authors for their replies to my questions and for their efforts to improve the manuscript.

Still regarding the first question of my first report, I could not find in the revised main text any discussion of the conditions required to preserve the nodal line. The authors write in their reply that the nodal line is protected by the mirror symmetry σ_z . What is exactly σ_z ? In some parts of the text / supplementary materials it is used as the Pauli matrix. It seems, however, that this σ_z represents the operation $(x,y,z) \rightarrow (x,y,-z)$. Is this correct? This should be clarified in the main text. A formal proof that the nodal line is indeed preserved against any (weak) perturbation by this symmetry would make the manuscript much stronger.

I maintain my original reservations about the suitability of this work for a high-impact journal. I do not see anything striking/new about the optical phenomena enabled by the "nodal ring": negative refraction can be obtained with much less complex designs (e.g., hyperbolic meta-materials), and similar scattering resonances can be found in many other optical systems.

Reviewer #3 (Remarks to the Author):

I found that the revised manuscript satisfies all my concerns with the original submission. The authors have briefly stated the unique properties of their system compared with condense matter and other optical NL works, seriously considered the indeed nonlocality of the unit cell, redone another more convinced experiment, shown more experiment details, and exactly replied for other minor comments. The current version is of high scientific quality and well written. Given the broad interest of the subject and the rapid progress in the field, I highly recommend this manuscript to be published in Nature Communications as is.

Reviewer #1 (Remarks to the Author):

I thank the authors for their comprehensive response. The finite element simulations of the eigenmodes convincingly demonstrate that the nodal-line degeneracy is not lifted by losses, and the experimental results are clearer with the use of the new lower loss structure. Therefore I think the major claims of the paper are now adequately supported. Before publication the authors should carefully check the text to ensure details are updated to reflect the new structure, e.g. line 101 refers to the previous nodal line frequency of 12.5GHz.

Reply: We thank reviewer for recommending our paper for publication and for pointing out the minor inconsistency in the text, which is now fixed in the revised manuscript.

Reviewer #2 (Remarks to the Author):

I thank the authors for their replies to my questions and for their efforts to improve the manuscript.

Still regarding the first question of my first report, I could not find in the revised

main text any discussion of the conditions required to preserve the nodal line. The authors write in their reply that the nodal line is protected by the mirror symmetry σ_z . What is exactly σ_z ? In some parts of the text / supplementary materials it is used as the Pauli matrix. It seems, however, that this σ_z represents the operation $(x,y,z) \rightarrow (x,y,-z)$. Is this correct? This should be clarified in the main text. A formal proof that the nodal line is indeed preserved against any (weak) perturbation by this symmetry would make the manuscript much stronger.

I maintain my original reservations about the suitability of this work for a high-impact journal. I do not see anything striking/new about the optical phenomena enabled by the "nodal ring": negative refraction can be obtained with much less complex designs (e.g., hyperbolic meta-materials), and similar scattering resonances can be found in many other optical systems.

Reply:

We thank reviewer for the comments. In our last reply σ_z represents the operation $(x,y,z) \rightarrow (x,y,-z)$. We add more details in the Supplementary Material (Note 6) to show that the nodal line is stable under mirror-symmetry preserving perturbations. Meanwhile we would like to point out that perturbations that break the mirror symmetry may gap out the nodal line. Finally, we would like to emphasize that negative refraction is only one of many

interesting optical phenomena that could result from the unique isosurface of the nodal line metamaterials. More importantly, the nodal line metamaterials, at the junction between different topological states, provide a unique platform for exploring topological phase transitions.

Reviewer #3 (Remarks to the Author):

I found that the revised manuscript satisfies all my concerns with the original submission. The authors have briefly stated the unique properties of their system compared with condense matter and other optical NL works, seriously considered the indeed nonlocality of the unit cell, redone another more convinced experiment, shown more experiment details, and exactly replied for other minor comments. The current version is of high scientific quality and well written. Given the broad interest of the subject and the rapid progress in the field, I highly recommend this manuscript to be published in Nature Communications as is.

Reply:

We thank reviewer for recommending our work for publication in Nature Communication.